# Development of a knowledge translation platform for ataxia: Impact on readers and volunteer contributors

**Celeste Elisabeth Suart**[1], **Katherine Jean Graham**[1], **Theresa Nowlan Suart**[2], **Ray Truant**[1]*

1 Department of Biochemistry and Biomedical Sciences, McMaster University, Hamilton, Ontario, Canada,
2 School of Medicine and Faculty of Education, Queen's University, Kingston, Ontario, Canada

* truantr@mcmaster.ca

## Abstract

### Background

Dissemination of accurate health research information to patients and families has become increasingly important with the rise of the internet as a means of finding health information. However, the public faces several barriers to accessing research information, including pay-walls and technical jargon. One method to bridge this gap between patients, families, and research is using lay summaries. SCAsource is an online knowledge translation platform where peer-reviewed research papers on ataxia are translated into lay summaries. This online platform was launched in September 2018, with the goal of making ataxia research more accessible and understandable to patients and families. A secondary goal is to provide opportunities for ataxia researchers to develop and hone their knowledge translation skills, altogether improving the quality of patient communication in the ataxia community.

### Aim

The aim of this study was to measure the impact of SCAsource on its readers and volunteer contributors after one year of activity. This is to ensure SCAsource is meeting its goals of (1) improving access and understanding of ataxia research to lay audiences, and (2) improving knowledge translation skills of volunteer contributors.

### Methods

Two online surveys were launched, one for readers and one for volunteers. Each survey had a combination of multiple-choice, Likert-scale type, and open-ended short-answer questions. Descriptive quantitative analysis was used for respondent characteristics and Likert-type data. A grounded theory coding approach was used to analyze narrative feedback data.

### Results

We found that SCAsource has mutually beneficial outcomes for both lay person readers and volunteer contributors. Readers have an increased understanding of ataxia research

**Data Availability Statement:** All relevant data are within the paper and its Supporting Information files.

**Funding:** This study was funded by the Krembil Foundation. The funders had no role in study design, data collection and analysis, decision to publish, or preparation of the manuscript.

**Competing interests:** The authors have declared that no competing interests exist.

and access to up-to-date information on recent publications. Volunteers develop knowledge translation skills and have increased confidence in communicating results to lay audiences. Areas of improvement were identified to be incorporated into the platform.

## Conclusion

We demonstrated that SCAsource improves access to information and understanding of research to lay audiences, while providing opportunities for researchers to develop knowledge translation skills. This framework can potentially be used by other rare disease organizations to launch and evaluate their own knowledge translation websites.

## Introduction

Disseminating research knowledge from academia to the general public has become increasingly stressed as an important activity [1, 2]. Knowledge translation, also referred to as knowledge mobilization or knowledge dissemination, is the practice of bridging this gap by making knowledge understandable and accessible for users [3, 4]. The Canadian Institutes of Health Research specifically stress the "synthesis, dissemination, exchange and ethically sound application" as key components of the knowledge translation process [5]. One popular knowledge translation model, Knowledge-to-Action, outlines both cycles of knowledge creation and knowledge application [5, 6] cation cycle involves adapting knowledge to the local context of users, as well as identifying barriers to using and accessing this knowledge [5]. End users include a variety of individuals, including health care professionals, policy makers, patients, and families [5]. As more members of the public use the internet as a means of accessing health information [7], the synthesis and dissemination of research knowledge to lay audiences is becoming a key responsibility of researchers, not merely an occasional by-product.

There are several barriers facing laypersons trying to access research information online. Often laypersons run into paywalls when trying to access primary research [8, 9]. When they are able to read articles through open access or subscriptions, then issues arise of highly technical language, scientific jargon, and impersonal writing style [9]. Although these stylistic choices are appropriate and even encouraged in academia, it can be alienating for lay audiences [9, 10].

Lay summaries have been demonstrated to make findings accessible and understandable to these non-specialist audiences [9–12]. This style of writing focuses on clear, engaging, and concise writing with the removal of technical jargon [13]. Despite the clear benefit of plain language summaries to lay audiences, many scientists struggle to write effective lay summaries [12, 14]. This difficulty is caused by many factors, including the vast difference in style between scientific and lay writing, the overabundance of scientific jargon, the heterogeneous nature of the lay audience, and fear of over-generalizing research findings [15].

One platform that has made extensive use of lay summaries is HDBuzz, an online knowledge translation website that focuses on Huntington's disease research [16]. Huntington's disease is a fatal neurodegenerative disorder caused by an abnormal expansion of CAG triplet repeats in the huntingtin gene [17]. HDBuzz was launched in January 2011 by Drs. Ed Wild and Jeffrey Carroll, motivated by discussions with Huntington's disease patients [16]. This platform provides short lay summaries written by clinicians or scientists, explaining how a particular research article fits into the broader Huntington's disease literature [16].

Inspired by HDBuzz and discussions with ataxia patients and family members at the 2018 National Ataxia Foundation's Ataxia Investigators Meeting, we wanted to launch a knowledge translation website focusing on another form of fatal neurodegenerative disease: Spinocerebellar ataxia (SCA). Lack of communication between ataxia researchers and patients has previously been identified as a barrier to patient engagement [18]. SCAs are a group of autosomal dominant disorders that primarily cause ataxia, the loss of motor control and balance [19]. Six subtypes of SCA are CAG triplet repeat expansion diseases like Huntington's disease, and there is some similarity in symptoms between these conditions [20].

In September 2018 we launched SCAsource, an online knowledge translation platform where peer-reviewed research papers on ataxia are translated into lay summaries. The main objective of SCAsource is to make ataxia research more accessible and understandable to patients and families. Secondary objectives include providing opportunities for junior ataxia researchers to develop and hone their knowledge translation skills, improving the quality of patient communication across the ataxia community.

SCAsource began as a low-budget pilot project, with initial start-up costs ($\leq$\$500 USD) being covered by members of the SCAsource team. The website was set up through WordPress, an online content management system which allowed for the creation of a professional website by persons with limited web design experience. We chose to create an independent website, as opposed to going through an already existing ataxia organization, to limit perceived bias towards a particular geographic location and reach as wide of an audience as possible. Article style and quality assurance guidelines were developed by volunteers who had previous knowledge translation training with other organizations. All content was licensed under a Creative Commons Attribution-ShareAlike 3.0 Unported License, to allow for dissemination on other websites. Initial advertisement of articles was done through social media with support from the National Ataxia Foundation.

Volunteer contributors are recruited through word-of-mouth, primarily at international conferences which focus on ataxia research. We have also had contributors contact SCAsource directly or be referred by current volunteers. They are mainly early career researchers from Canada, the United States, and Europe. Most volunteers come from a basic science background. However, more clinical researchers have signed up to write for SCAsource as it has expanded into covering clinical trial results. New volunteers are giving a training guide on how to write effective lay summaries, providing constructive editing feedback, and document guidelines for our two specific article types. These guidelines also provide an overview of minimum quality standards required for publication, timelines for writing articles, and a brief introduction to the knowledge translation literature. These new writers are then paired with more experienced editors during their first few volunteer experiences, to allow for mentorship on knowledge translation to occur through the writing and editing process.

Currently, SCAsource has two regularly updated article types (Summaries and "Snapshots") and two "static" reference resources (a glossary and introduction to Ataxia article). SCAsource Summaries convey the findings and implications of entire research articles, as well as the context in which these discoveries were made. The Summary article type was modeled on the lay article style successfully used by HDBuzz [16]. Summaries follow the inverted pyramid structure and best lay summary practices described by Salita [15]. SCAsource Snapshots focus on discrete scientific concepts and background knowledge. The Snapshot article type was launched in April 2019 in response to early reader feedback requesting a deeper explanation of core concepts that appear in multiple SCAsource Summaries. They follow the best practices derived from COGFAST, where members of the public were consulted on what format and content they prefer in lay summaries [12]. Topics for Snapshots are generated from search terminology which brings readers to the website or through social media discussions. All

SCAsource content is published under a Creative Commons license, making it freely available to distribute.

SCAsource contributors follow a month-long article writing and editing process. At the beginning of each month, a list of research articles for Summaries and topics for Snapshots is circulated amongst contributors. These lists are compiled through a combination of suggestions by email and social media, recently published articles, and search engine information which directs readers to SCAsource. Writers have two weeks to create first drafts, followed by one week of editing by a second contributor, after which they have one week to submit revised articles. Articles can be flagged for a second or third round of editing if further improvements are required. All articles are sent through a copy-editing process to ensure they meet minimum publication standards. This includes a suitability for general audience score of 80% using De-Jargonizer, an automated jargon identification program [21].

In September 2019, we launched an online survey to determine if SCAsource was meeting its mandate objectives of improving readers' knowledge of ataxia research and volunteers' knowledge translation skill sets. The objective of this study was to determine the impact of SCAsource on its readers and volunteers, establish strengths of the platform, and identify areas of improvement. Through this study, we hope to provide a framework for which other disease groups can launch and evaluate their own low-initial-cost knowledge translation websites.

## Methods

### Ethics approval

This study was evaluated by the Hamilton Integrated Research Ethics Board (Project Numbers 7425 & 7426) and determined to be exempted from ethics review due to it being considered secondary use of anonymous quality assurance data.

### Study design, participants and recruitment

Two parallel online surveys were launched from September 27, 2019 to December 2, 2019; one for SCAsource volunteer contributors and one for SCAsource readers. Both surveys were administered through the LimeSurvey platform, taking approximately 20–30 minutes to complete. The surveys comprised of Likert-scale and multiple-choice type quantitative questions, along with open-ended qualitative questions.

No financial incentive was given for either survey. To increase the response rate, a follow-up email was sent two weeks after initial contact.

Thirty-three SCAsource volunteers met the selection criteria for the contributor survey. This included (i) having written or edited at least one article for SCAsource between September 2018-September 2019, and (ii) not being an investigator on this study. Potential respondents were contacted by email through the SCAsource volunteer email list. They were given a letter of information about the study and a link to the online survey.

Our inclusion criteria for the reader survey were individuals who (i) had read at least one SCAsource article between September 2018-September 2019, (ii) were 16 years of age or older, (iii) did not act as a contributor to SCAsource, and (iv) were not an investigator on this study. Estimating the population size eligible for the reader survey was more challenging, as visitor information to the website is measured in IP address statistics. More than one individual could use the same IP address, or one person could use multiple IP addresses. To recruit readers, an email was sent to the SCAsource subscription list (57 eligible participants) including the study letter of information and link to the survey. Two pinned posts advertising the survey were published on the SCAsource website and Twitter account to engage readers who visit the website but are not subscribed for updates.

## Analysis

Once data was collected, survey response data was formatted and transferred to the qualitative data analysis software MAXQDA (VERBI GmbH, Berlin, Germany). Descriptive statistics were generated for both volunteer and reader surveys. Website visit data was obtained using the WordPress Jetpack plugin. Quantitative data was entered into GraphPad Prism 8 for analysis and formatting.

To analyze qualitative data, we took a social constructivist approach to grounded theory as described by Charmaz for open coding [22]. Two researchers independently completed thematic analysis following a *line-by-line* open coding approach in MAXQDA [22]. These initial codes were then synthesized into key categories by identifying interrelated concepts. All codes were reviewed for agreement, with discrepancies resolved through discussion until consensus was reached. This master coding list was given to a third independent researcher to see if the themes previously identified would be subsequently identified by an individual who had not previously worked with the data.

We ensured the rigour of the qualitative analysis by embedding strategies outlined by Lincoln and Guba's four criteria of rigour: credibility, dependability, confirmability, and transferability [23, 24]. For credibility, we surveyed both contributor and reader populations to ensure a holistic view of SCAsource, along with ensuring investigators had the required knowledge of the website and qualitative coding methodology. For dependability and confirmability, we have included a detailed description of the coding process used with multiple independent coders, which included investigator triangulation to remove potential vias from the analysis. For transferability, we had a clear description of the research context and assumptions.

## Results

### SCAsource website performance

As of January 2020, SCAsource has had over 26,900 views from over 124 countries (Fig 1). Website views display seasonal fluctuations, receiving fewer views during winter holidays, with an overall increasing trend between years (Fig 1A). Over half of SCAsource website views originate from the United States, followed by Canada (10%), the United Kingdom (5.8%), China (2.8%), and India (2.2%) (Fig 1B).

### Respondent sample characteristics

We had an overall response rate of 58% (19/33) for volunteers, which is higher than most e-mail survey response rates [25]. Of the volunteers who responded to the survey, 74% (14/19) completed all sections, while 32% (6/19) skipped the qualitative feedback portion. The volunteer demographic information is summarized in Table 1. Most respondents were either graduate students (32%, 6/19) or postdoctoral researchers (37%, 7/19). Over half of respondents contributed two to three articles to SCAsource between September 2018 to September 2019. Most volunteers report they read SCAsource content, with 68% (13/19) visiting the website once a month.

We had 36 respondents to the reader survey, with 75% (27/36) completing all sections of the survey. Although we initially hoped for a greater response rate, this level of participation is not surprising as one symptom people in our target demographic may experience is difficulty with fine motor tasks. This was highlighted in the quote, "typing is hard" by Reader 18. This barrier inherent to the use of an online survey protocol may explain the reduced rates of response. A more accessible alternative would be to conduct in-person, semi-structured interviews. This approach was not feasible for this study because of the worldwide distribution of SCAsource readers and limited research funding. The demographic information of the reader respondent sample is summarized in Table 2.

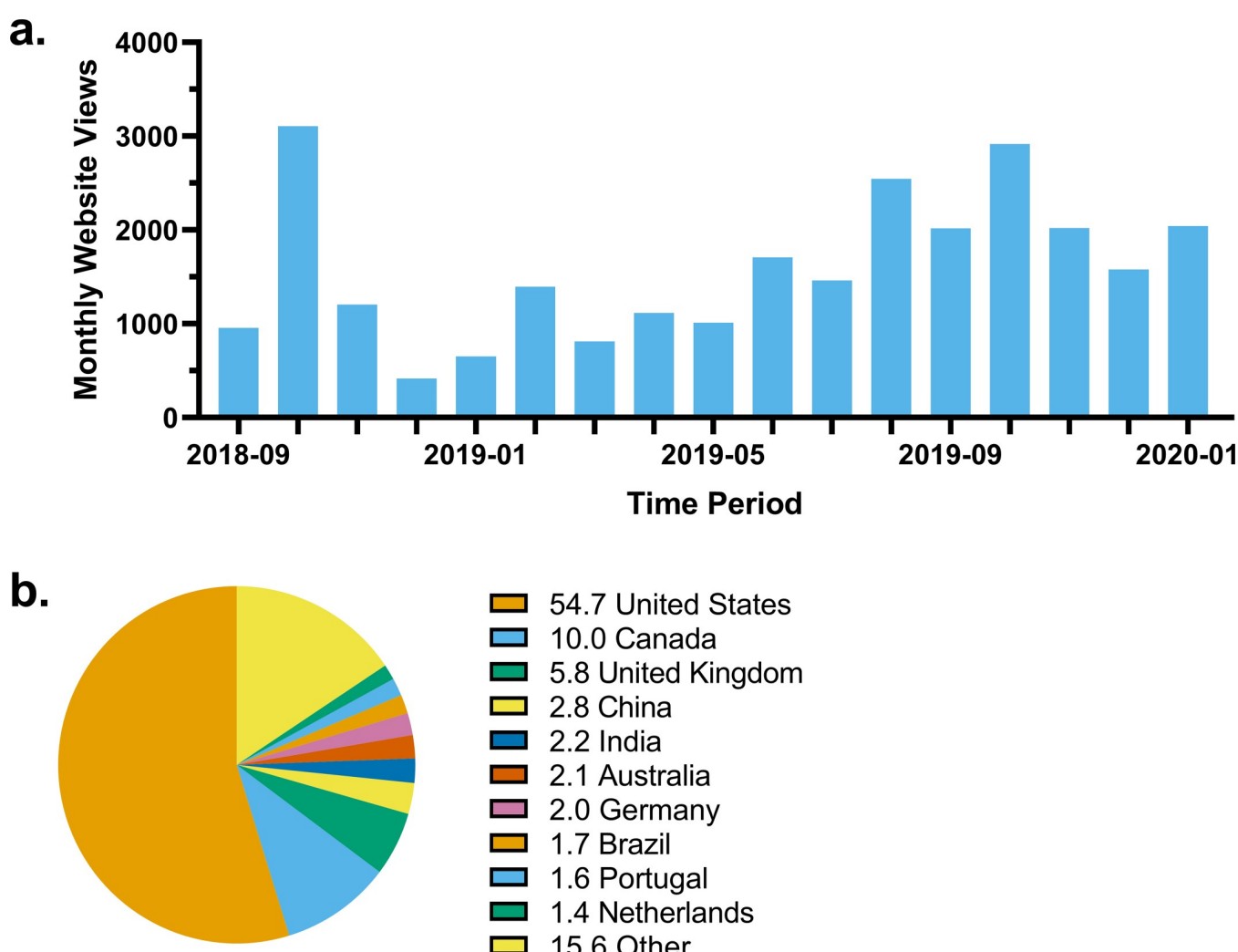

**Fig 1. SCAsource website visit statistics.** (A) Total website views of SCAsource.net per month. Time period ranging from September 2018 to January 2020. (B) Country of origin of SCAsource viewers. Top ten individual values displayed with representative percentages.

Over half of reader respondents (53%, n = 19) had read over seven SCAsource articles between September 2018 to September 2019. This high level of engagement may be explained by our recruitment using the SCAsource subscription email list, as 39% (14/36) of respondents reporting using the subscription list. Readers reported frequently searching for ataxia information, with 82% (29/36) searching online once a month or more. The sources that readers reported using most frequently were the National Ataxia Foundation (64%, n = 23), an American ataxia charity located in Minnesota, and search engine results (61%, n = 61). The SCAsource website was the third most used source of ataxia information at 56% (n = 20). When asked where they find out about SCAsource content, the top sources cited by readers were the SCAsource website (39%, n = 14), SCAsource subscription list (39%, n = 14), the National Ataxia Foundation's social media (36%, n = 13), and search engine results (28%, n = 10).

### Impact of contributing to SCAsource on volunteers

Feedback from volunteers on the impact of SCAsource on their skill development was generally positive, as depicted in Fig 2. Over half the volunteers (58%, n = 11) agreed that

**Table 1. Volunteer respondent characteristics.**

| Characteristic | N (%) |
|---|---|
| Position | |
| Graduate Student | 6 (32) |
| Postdoctoral Researcher or Fellow (PDF) | 7 (37) |
| Principal Investigator | 2 (10) |
| Other* | 4 (20) |
| Articles Contributed to SCAsource | |
| 1 | 3 (16) |
| 2 to 3 | 11 (58) |
| 4 to 5 | 2 (10) |
| 6 or more | 3 (16) |
| Readership of SCAsource | |
| Yes | 16 (84) |
| Yes, but only articles to which they contributed | 3 (16) |
| No | 0 (0) |
| Frequency of Reading SCAsource | |
| Once every few months | 4 (21) |
| Once of month | 13 (68) |
| Once a week | 2 (10) |

*Examples of "Other" category positions included research technician, consulting scientist, and medical writer.

contributing to SCAsource improved their writing or editing skills, with 90% (n = 32) saying it improved their confidence when communicating to lay audiences (Fig 2). Volunteers also reported the amount of time they dedicate to knowledge translation activities increased (74% agree or strongly agree, n = 14), although the majority rated it had no impact on their time management skills (63% neutral, n = 12) (Fig 2). Sixty-two percent (12/19) agreed that volunteering for SCAsource was beneficial to their development as a scientist, with 53% (10/19) stating the experience enhanced their understanding of ataxia literature (Fig 2). Most respondents (89%, n = 17) agreed they saw volunteering for SCAsource as a way to give back to the ataxia patient community (Fig 2). Based on these quantitative measures, SCAsource volunteers reported a gain in knowledge translation skills, including writing, editing, and lay audience communication, in addition to increased time spent on knowledge translation activities.

Similar themes of skill development and confidence in knowledge translation also emerged from the analysis of qualitative responses. Table 3 outlines the key themes identified from volunteer narrative data, along with representative quotations.

Volunteers reported that their experience with SCAsource changed their writing style when communicating with a lay audience. This includes how they structure information, their use of understandable terminology, and identifying key takeaway messages from research articles (Table 3). For some, contributing to SCAsource made them intentionally self-reflect during the lay summary writing process. As one volunteer explained, contributing to SCAsource "forced me to slow down and consciously question my word selections," (Volunteer 1, Principal Investigator). As it has been previously shown that researchers struggle with choosing appropriate lay terminology [14], hence, our volunteers reporting this level of conscious awareness of word choice is promising.

Multiple volunteers also identified improved confidence in knowledge translation as the main impact SCAsource has had on them, mirroring the quantitative Likert-style data (Fig 2). This is likely tied to the high proportion of volunteers for whom SCAsource was one of their

**Table 2. Reader respondent characteristics.**

| Characteristic | N (%) |
|---|---|
| **SCAsource Articles Read** | |
| 1 to 2 | 4 (11) |
| 3 to 4 | 4 (11) |
| 5 to 6 | 9 (25) |
| 7 or more | 19 (53) |
| **Frequency of Searching for Ataxia Information** | |
| Less than once a year | 1 (3) |
| Once every few months | 6 (17) |
| Once a month | 11 (31) |
| Once a week | 11 (31) |
| More than once a week | 7 (20) |
| **Source of Ataxia Information** | |
| SCAsource Website | 20 (56) |
| Search Engine (Google, Bing, Etc.) | 22 (61) |
| National Ataxia Foundation | 23 (64) |
| Social Media (Facebook, Twitter, Instagram) | 5 (14) |
| Shared Friends & Family | 1 (3) |
| Other** | 5 (14) |
| **Source of SCAsource Article** | |
| SCAsource Website (Direct Visit) | 14 (39) |
| SCAsource Subscription Email List | 14 (39) |
| Search Engine Result | 10 (28) |
| National Ataxia Foundation (Social Media) | 13 (36) |
| Social Media (Facebook or Twitter) | 4 (11) |
| Shared Friends & Family | 2 (6) |

**Examples of "Other" category sources of ataxia information European ataxia organizations, social media, and health news platforms.

first opportunities to engage in knowledge translation. As described by Volunteer 12, "I'm not used to communicating others' results to lay audiences and this has allowed me to practice". This "opportunity for practice" (Volunteer 9, Graduate Student) was highlighted as a main strength of the SCAsource initiative overall.

The other key strength of SCAsource from the perspective of volunteers was the potential utility to patients and families. Volunteers specifically liked the breadth of topics covered and the emphasis of current research being quickly communicated (Table 3).

When asked about potential areas of improvement, contributors identified training for new volunteers and public awareness of SCAsource. Currently, when new volunteers are onboarded, they are given three documents outlining the SCAsource guidelines on summary writing, Snapshot writing, and editing. In total there are six pages of readings, with additional suggested readings for those interested. Volunteers suggested this training could be more engaging, such as a video or web module (Table 3). Volunteers also pinpointed visibility and general awareness as an area of improvement (Table 3). This is consistent with informal feedback received when new contributors contact the SCAsource executive. Increased social media use was suggested as a potential solution.

Volunteers also expressed that the general concept of SCAsource was a good idea and gave encouragement for the initiative to continue (Table 3). Some volunteers also expressed being

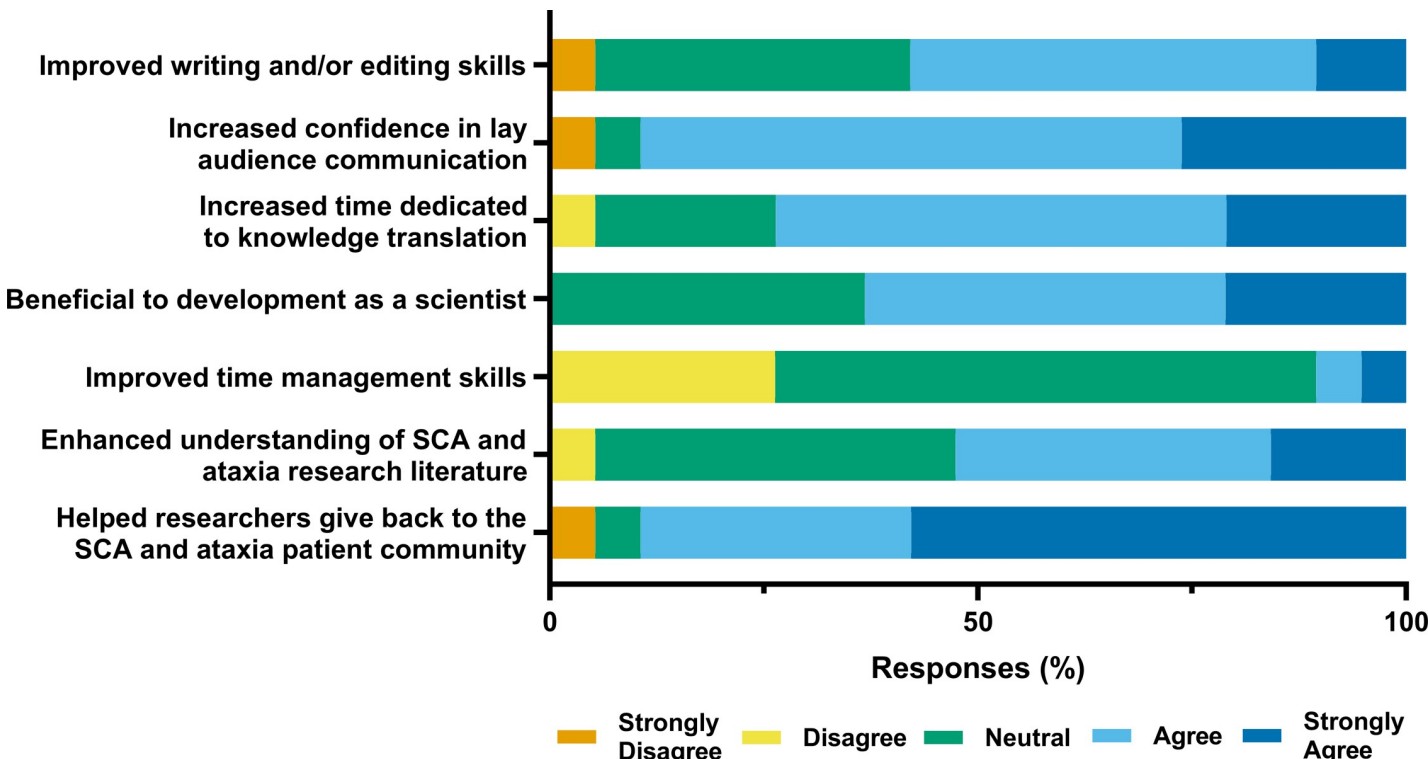

**Fig 2. Contributors self-reported outcomes of volunteering for SCAsource.** Respondents were asked to rate their agreement about whether volunteering for SCAsource resulted in the above statements using the indicated 5-point Likert-type scale.

"grateful for the opportunity to contribute" (Volunteer 7, PDF) to SCAsource. This reflects the overall positive impression that contributors have of SCAsource, both with regards to personal skills development, increased confidence, and being able to make an impact on the SCA community.

## Impact of SCAsource content on readers

Readers reported an overall positive effect of reading SCAsource content (Fig 3). Over 88% (n = 32) agreed that reading SCAsource increased their understanding of ataxia research, while 83% (n = 30) reported they have learned more about ataxia (Fig 3). When asked if SCA-source helped them feel more connected to ongoing ataxia research, 86% (n = 31) of respondents agreed (Fig 3). A majority (94%, n = 34) reported trusting SCAsource as an unbiased source of information (Fig 3). Responses were more varied when polled about how SCAsource influenced their interest in participating in current ataxia research or clinical trials. Sixty-one percent (22/36) agreed that reading SCAsource had increased their interest in participating in such studies, while 39% (n = 14) were neutral on the subject (Fig 3).

Readers were then asked to rate how helpful they found four different types of content on SCAsource; Summaries, Snapshots, the glossary, and the "What is Ataxia?" information page (Fig 4). Summary and Snapshot articles represent the majority of SCAsource content. Summaries are longer texts (800–1000 words) where scientific research is summarized and reported on. SCAsource snapshots are 400 words or less, focus on explaining one scientific topic clearly and concisely. The glossary and "What is Ataxia?" information page are static pages on the SCAsource platform that are infrequently updated. The glossary contains commonly used

**Table 3.  Volunteer themes and representative quotations.**

| **Knowledge Translation Skill Development** | |
| --- | --- |
| **Improved communication of scientific findings to lay audiences through knowledge translation techniques** | "It is very important that scientists explain lab findings to the general public and (even more important) the SCA patients. Volunteering helps to explain difficult scientific terms to easier and understandable terms." (Volunteer 10, PDF) |
| | "I'm not used to communicating others' results to lay audiences and this has allowed me to practice extracting the key findings in papers and presenting them in a meaningful and easy way. It has also given me experience in non 'scientific' writing." (Volunteer 12, Graduate Student) |
| | "Volunteering for SCAsource has improved how I frame the information I want to communicate to the general [public]. Before SCAsource, I didn't realise that information needed to be presented in a different order and structure to ensure maximal understanding by the general public." (Volunteer 7, PDF) |
| **Connection and confidence in communication with ataxia patients** | "Writing articles for SCAsource has helped me put myself in the shoes of ataxia patients. I have been able to better empathize with patients by thinking about articles, how they relate to the situations of ataxia patients, and why patients should care about scientific research. This makes it easier to not only write future articles, but also help me get practice for how to speak with other lay members about science." (Volunteer 3, PDF) |
| | "[Volunteering] has given me more confidence in my ability to communicate to a lay audience through writing." (Volunteer 18, Consulting Scientist) |
| | "[Volunteering] has helped with my confidence in communicating clearly in plain language." (Volunteer 5, Research Technician) |
| **Strengths of the SCAsource Initiative** | |
| **Provides opportunity for researchers to practice knowledge translation** | "It has also done a great job of allowing many members of the ataxia research community to get involved." (Volunteer 3, PDF) |
| | "[SCAsource has] Provided many opportunities for practice." (Volunteer 9, Graduate Student) |
| **Comprehensive platform for laypersons to learn about current research ataxia** | "I think that having the mix of articles, snapshots and the glossary is great. I think together they all provide a really comprehensive platform for the general public to learn about the current research and background of SCAs." (Volunteer 7, PDF) |
| | "I think SCAsource has done a good job at covering current topics and at providing a good platform for scientists to communicate with the SCA community." (Volunteer 18, Consulting Scientist) |
| **Areas of improvement for the SCAsource Initiative** | |
| **More extensive volunteer training** | "I think a training module, either a short video or a slide deck, could be made to explain how to properly write a summary for a lay audience." (Volunteer 3, PDF) |
| | "I also think more training or guidance should be provided on how to communicate information to patients. As researchers we are used to 'overselling' the translational impact of our research for grant applications and it can be difficult to change tone and communicate to patients in a way that doesn't give false hope or isn't mistaken for medical advice. As scientists I think we need guidance on how to realistically communicate science to patients." (Volunteer 6, Graduate Student) |

*(Continued)*

**Table 3.** (Continued)

| Need to improve awareness and visibility | "Visibility. I've only heard about SCAsource through my collaboration." (Volunteer 17, Other) |
|---|---|
| | "As SCAsource gains more traction (and philanthropic funding) it would be interesting to explore reporting summaries from NAF conferences or have a social media presence toward in press articles/what is coming down the pipeline." (Volunteer 1, Principal Investigator) |
| | "Better PR: linkedin page, facebook etc." (Volunteer 4, PDF) |
| **Positive feelings about participating in the SCAsource initiative** | |
| | "I really enjoy being part of the SCAsource community and think that it is a great platform for the general public to learn about SCAs. I am very grateful for the opportunity to contribute to this effort." (Volunteer 7, PDF) |
| | "I think what [SCAsource is] doing is great and deserves more traction." (Volunteer 12, Graduate Student) |
| | "Very valuable contribution to the community!" (Volunteer 16, Principle Investigator) |
| | "Keep up the good work!" (Volunteer 18, Consulting Scientist) |

words across all article types. The "What is Ataxia?" page is aimed at readers who are new to ataxia and is written as a general overview of ataxia information covered on the website.

Both SCAsource Summaries and Snapshots were the content types rated most helpful by readers (Fig 4). A third of respondents (11/33) classified Summaries as extremely helpful, while 42% (14/33) ranked Summaries as very helpful (Fig 4). ewer readers rated Snapshots as

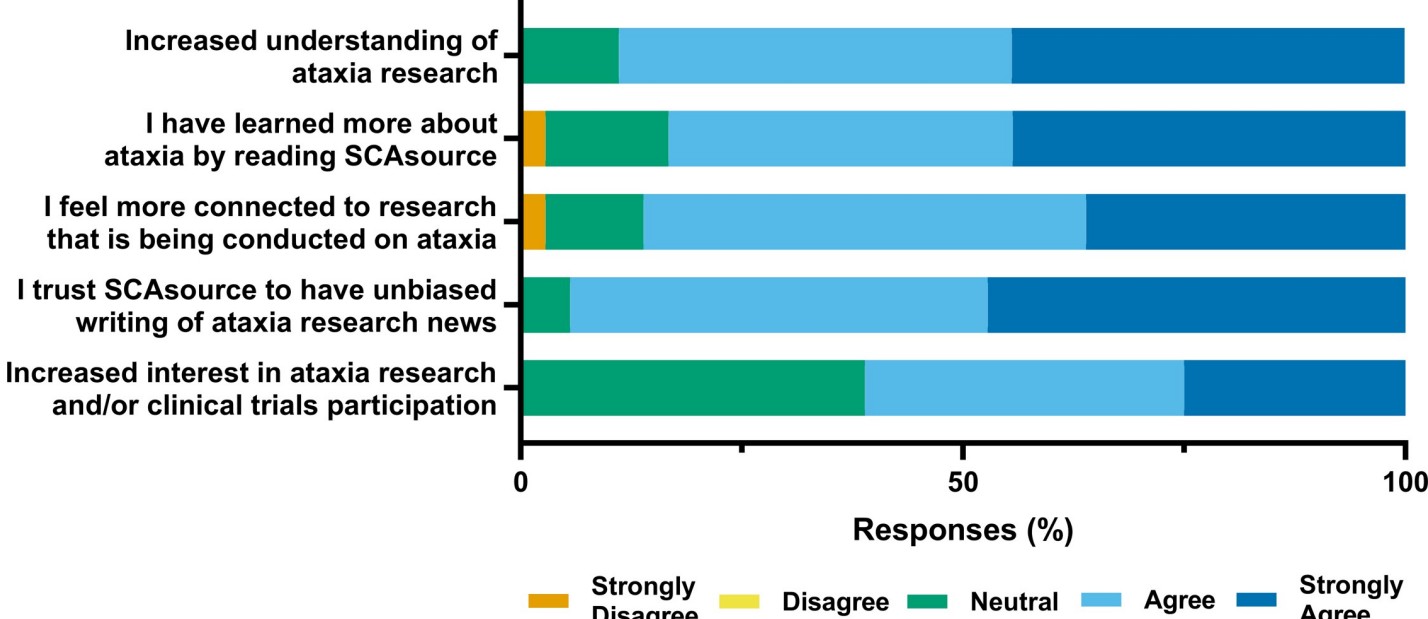

**Fig 3. SCAsource reader self-reported outcomes.** Respondents were asked to rate their agreement about whether reading SCAsource content resulted in the above statements using the indicated 5-point Likert-type scale.

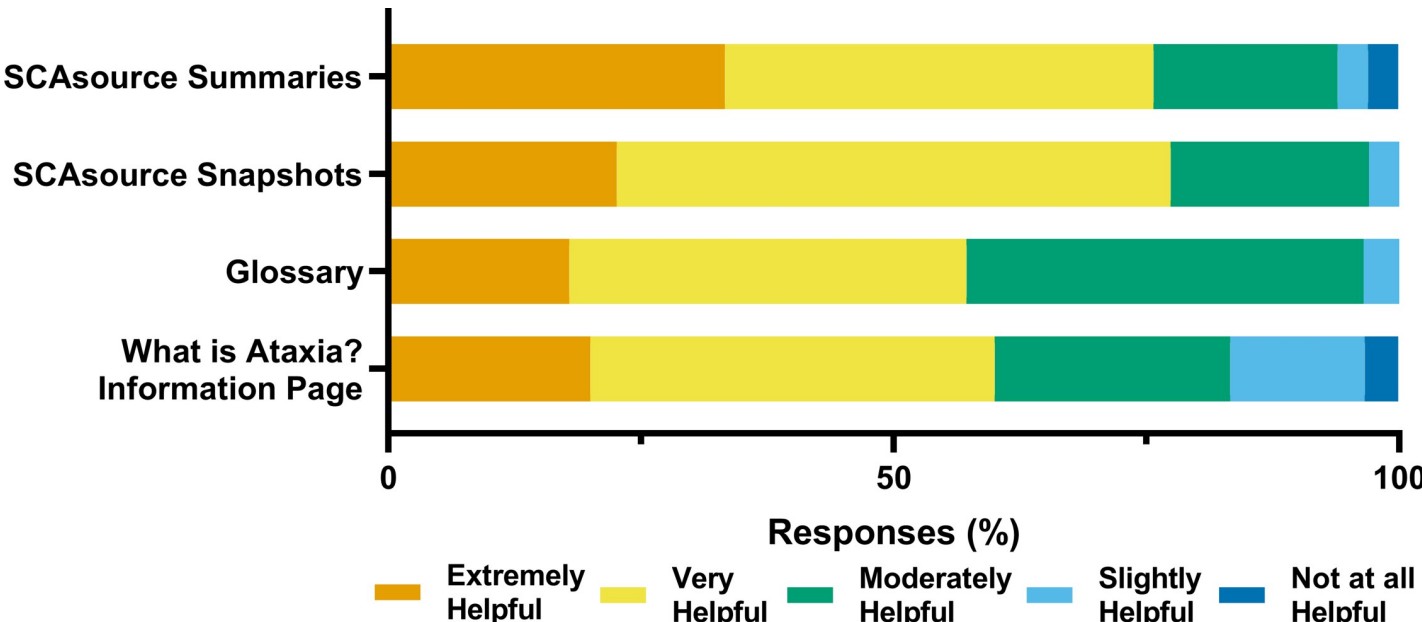

**Fig 4. Reader helpfulness ratings of SCAsource content.** Respondents were asked to rate the helpfulness of four content types of SCAsource; summaries, snapshots, the glossary, and the "What is Ataxia?" information page. Responses were given with the indicated 5-point Likert-type scale.

extremely helpful (23%, 8/33) compared to summaries (Fig 4). However, over half of readers (55%, 18/33) reported Snapshots as very helpful (Fig 4). Both static pages had lower helpfulness from readers. The glossary ranked extremely or very helpful by 57% (19/33) of readers, with 43% (14/33) classifying it as moderately or slightly helpful (Fig 4). The "What is ataxia?" page had more variability in responses. Although 60% (20/33) described this content as extremely or very helpful, the remaining 40% (13/33) of readers described it as moderately, slightly, or not at all helpful. Overall, readers viewed frequently updated content, such as Summaries and Snapshots, as more helpful to them compared to static content on SCAsource.

Through analysis of narrative data, we took a closer look at what exactly readers found helpful about SCAsource content. Themes that emerged included an emphasis on clarity and access to information, as well as suggestions to improve the SCAsource initiative. Key themes from SCAsource readers, along with representative quotations, are outlined in Table 4.

Most reader respondents appreciated the easy-to-understand content, that SCAsource is an accessible resource, and that SCAsource provides information about ongoing research (Table 4). As Reader 7 explained, "The articles are easier to understand than most ataxia articles". A few also mentioned they like how SCAsource provided links to the original research, as well as additional resources, so that they could explore topics further. This mirrors past findings highlighting patient interest in primary scientific literature [9]. There was also an emphasis on "up to date" (Reader 25) research and being able to see progress being made. Readers' motivation for their interest in SCAsource differed–from understanding their own condition, a child's, or a friend's.

A variety of improvements for SCAsource were suggested, with the theme of current and ongoing research again emerging (Table 4). Readers requested more information on research they could participate in, what research questions are being investigated, where research is taking place, and who are the scientists doing this work. A handful of readers requested more frequent updates to the website, again tying into this idea of receiving the latest updates. Like SCAsource volunteers, readers also identified advertising and communication as an area of

**Table 4. Reader themes and representative quotations.**

| | |
|---|---|
| **Strengths of the SCAsource Initiative** | |
| **Easy to understand content** | "Scientific research written in a way that is clearly understandable." (Reader 29) |
| | "I really like that difficult topics for non-scientists to understand, such as RAN translation, is explained in a more accessible way to patients." (Reader 36) |
| | "Easily written but not to short and not to simplified" (Reader 8) |
| **Accessible resource and information** | "Quicker access to information regarding SCA and being able to link to other sites and resources for additional information. Not being a researcher myself, the information is produced in understandable language for the average person." (Reader 10) |
| | "[SCAsource has] information regarding up to date research." (Reader 14) |
| | "Very good communication channel for ataxia research" (Reader 20) |
| | "It is an excellent resource" (Reader 29) |
| **Information on ongoing research** | "Good summaries, research articles / to see some progress in the research" (Reader 8) |
| | "I like to hear about the research that is currently ongoing." (Reader 3) |
| **Areas of Improvement for the SCAsource Initiative** | |
| **More information about research that readers can participate in** | "More information about research studies that people can participate in, and how they can participate." (Reader 36) |
| | "Some more information about the studies, e.g. READISCA, etc. Like an overview to get even more people involved" (Reader 8) |
| **More information on ongoing research, who is doing research** | "A round-up of EVERY current (and past) research project." (Reader 3) |
| | "Timeline of what is going on in the 'research' world" (Reader 8) |
| | "[I would like] pictures of the authors" (Reader 5) |
| **More frequent updates** | "Should be updated more frequently." (Reader 21) |
| **Better advertisement of content** | "The only time I see new articles is when random patients post them in the NAF facebook groups! You should have a facebook page" (Reader 36) |
| **Lack of suggestions for improvement** | "I don't know. Everything seems correct." (Reader 4) |
| | "Nothing—I think it is excellent" (Reader 29) |
| | "It's great" (Reader 12) |
| **Appreciation for SCAsource as a Resource** | |
| **Gratitude for the creation of SCAsource** | "Keep on keeping on. And remain upbeat about it." (Reader 3) |
| | "I really like to see that the content is posted regularly and I really hope that the site will stay." (Reader 8) |
| | "Thank you for your work." (Reader 33) |

improvement. In addition to the common themes for suggested improvement, some suggestions stemmed from individuals' personal preferences or needs, including a request for language translation and a request for promotion on a specific social media platform. While these will be considered in future plans for SCAsource, they will be lower priority items.

We were surprised by the number of reader respondents who advised that there were no areas of improvement for SCAsource (Table 4). When asked about how SCAsource could be improved, one reader answered "I don't know. Everything seems correct." (Reader 4). This supports that the SCAsource platform is currently working. Similar to how volunteers were grateful for participating in SCAsource, many readers also gave thanks for the creation of SCAsource. Readers expressed that it was an "excellent resource" (Reader 29) and asked that volunteers "Keep up the good work" (Reader 10).

## Discussions

In this study, we assessed the self-reported impact SCAsource has on its readers and volunteers. This was done through a mixed-methods analysis of online survey data from 36 readers and 19 volunteers. Overall, both groups reported a positive evaluation of SCAsource. We demonstrated that the dissemination strategy used by HDBuzz [16] can be modified successfully to serve other disease interest groups.

Over the past 17 months, SCAsource has overall observed an increasing trend of views (Fig 1A). Views display some seasonal trends, with lower views over the winter holidays, a pattern known to occur with platforms not selling commercial goods. SCAsource also had spikes in view count in both October 2018 and 2019 (Fig 1A). We postulate that could be in part due to increased sharing of SCAsource content following International Ataxia Awareness Day on September 25. Over 70% of SCAsource views originate from primarily English-speaking nations, with 64.7% coming from Canada and the United States (Fig 1B). This is reflective of the location of SCAsource contributors, who are primarily from laboratories located in North America. If SCAsource is to expand its readership to reach more ataxia patients and families, it will also need to expand its volunteer base to include more researchers from international laboratories. Volunteers with fluency in languages other than English will help expand SCAsource from a unilingual to a multilingual initiative, following a similar trajectory to HDBuzz [16].

Current volunteers reported a key strength of SCAsource was the opportunity to practice knowledge translation. This opportunity for practice and training is possibly what led to the self-reported gains in knowledge translation skills, as well as improved confidence in communicating with lay audiences. This suggests that SCAsource filled a gap in training for researchers, giving them a supportive environment with constructive feedback to improve their lay summary writing. This is further reflected by the request for more extensive knowledge translation training for volunteers.

In addition to an increased understanding of ataxia research, SCAsource readers reported they felt an increased connection to ongoing ataxia research through this platform. The theme of up to date, current research was present throughout multiple sections of the reader survey responses. Readers preferred SCAsource content which updated every week over static informational content. Access to information about ongoing ataxia research was cited as both a strength and a potential area of improvement. This indicates SCAsource is on the right track with regards to summarizing recently published research, but we could expand this area more. In response to this feedback, SCAsource is planning to launch a new article type that will give information about ataxia research laboratories. This will include where the laboratories are located and what areas of research they are pursuing. Our aim is that this new article type will meet the need of readers wanting to learn more about ataxia researchers, the research process, and ongoing studies.

Readers had more mixed responses with regards to whether reading SCAsource content increased their interest in participating in clinical trials (Fig 3). Past research on barriers to patient participation in clinical trials have identified lack of information and understanding as a barrier [26, 27]. Conversely, investigation into neurological clinical research participation has found that receiving information from a trusted source is a key motivator for patient participation [28]. SCAsource does not actively advertise clinical trial recruitment. Its focus is on explaining results from published clinical trial data, as well as clarifying methodological procedures common across trials. Despite this, over 60 percent of reader respondents indicated their interest in clinical trial participation has increased (Fig 3). However, it can be argued that patients who are already more interested in research participation could be more likely to seek out research information. For this reason we must be cautious not to draw direct causation

from these results. Further analysis with a larger respondent sample would be needed for future investigation into this topic.

Suggested areas of improvement from both volunteers and readers point to growth opportunities for SCAsource. This includes more frequent article updates and additional training for volunteers. This feedback points to a well-received knowledge translation website that has room to grow with additional financial support.

Themes from both surveys also demonstrate that this kind of knowledge translation platform can serve both the research community and the community of those affected by ataxia (patients, families, friends). Embedded in the feedback from both surveys is the respect and gratitude each community has for the other. There was no sense of imbalance, incorrect focus, or one community benefiting over the other. Early career researchers were able to practice valuable knowledge translation skills, while readers gain knowledge about ongoing ataxia research. This positions SCAsource as a mutually beneficial platform connecting research and lay ataxia communities.

We propose that knowledge translation set-ups like HDBuzz and SCAsource could be used by other disease organizations, especially those concerned with rare diseases. More common research areas, such as cancer, heart disease, and diabetes, tend to have more established knowledge translation initiatives [29–31]. Conversely, rare disease organizations have identified knowledge translation as a needed area of growth [32]. One potential barrier is cost, as rare diseases typically receive less research funding and overall investment due to impacting only a small portion of the population [33, 34]. The knowledge translation set-up we outline in this manuscript reduces barriers to entry due to its low cost. This would allow for organizations to begin a knowledge translation initiative and generate enough interest to attract external funding to continue and improve efforts over time.

Multiple knowledge translation models have been documented in the literature, with overlap allowing for key themes to emerge from the discipline [35]. Such themes include the importance of knowledge translation being ongoing interactive processes with stakeholders, and involving multiple people with varying perspectives about ongoing research [36, 37]. The Knowledge-to-Action model, emphasizes the recurrent nature of knowledge creation and implementation action cycles [5]. However, these existing models do not address specific issues that are relevant to the rare disease context [35].

Across knowledge translation models there is an emphasis on monitoring outcomes, including improved survival and quality of life [35]. However, the novel treatments for rare diseases, including ataxia, tend to have incremental effects over the span of years [35]. Another assumption by knowledge translation models is that the knowledge being translated has immediate clinical impact for the reader and their healthcare decisions. However, most of the research that SCAsource is asked to cover are in either pre-clinical or in early clinical trials. These discoveries do not have an immediate effect on the reader's care. Rather than traditional outcomes measured, learning more about these findings may influence a reader's choice to participate in research, or provide hope that new discoveries are being made.

## Study limitations

A limitation of this study was the use of a self-reported online survey format for gathering data. As previously discussed, the use of this method may have been a barrier to readers having trouble with typing and other fine motor tasks. A more accessible alternative for future work would be conducting verbal, semi-structured interviews, either in-person or online through video conferencing. We believe this is one likely reason the number of respondents to the reader survey was lower than we wanted. Using number of unique IP addresses visiting the SCAsource website between September 2018 and January 2020 (approximately 11,331) to

estimate total readership, the respondent sample represents less than 1% of total SCAsource readership. For this reason, we may not have reached saturation of themes from narrative data from readers. Our population sample may also have been biased towards more frequent readers of SCAsource. This is supported by over half of reader respondents reporting reading 7 or more SCAsource articles (Table 2). Thus, the views expressed by most reader respondents may not hold true for more casual readers of the website.

A second limitation is that our Likert-type scales focused on self-reported outcomes which were not objectively assessed through other means. Future work should include assessment of whether volunteer and reader self-reported gains align with gains measured through other objective means.

## Conclusions

We found that SCAsource has mutually beneficial outcomes for both lay person readers and volunteer contributors. Volunteers develop knowledge translation skills and have increased confidence in communicating results to lay audiences. Readers have an increased understanding of ataxia research and access to up to date information on recent publications. Areas of improvement were identified and will be worked towards to improve the SCAsource initiative. We build on past work by HDbuzz [16] to demonstrate this knowledge translation framework is effective in the context of other rare diseases. Further, we provide a foundation on which others can evaluate the effectiveness of their own knowledge translation websites.

## Supporting information

**S1 File.**
(DOCX)

## Acknowledgments

The authors would like to thank all individuals who responded to the volunteer and reader surveys. Many thanks as well to all current SCAsource volunteer writers and editors whose efforts made this website a reality. This work would not be possible without the support of both the ataxia researcher and patient communities.

Copies of SCAsource training materials and resources are available upon request by email (truantr@mcmaster.ca).

## Author Contributions

**Conceptualization:** Celeste Elisabeth Suart, Theresa Nowlan Suart, Ray Truant.

**Data curation:** Celeste Elisabeth Suart, Katherine Jean Graham, Theresa Nowlan Suart.

**Formal analysis:** Celeste Elisabeth Suart, Katherine Jean Graham, Theresa Nowlan Suart.

**Funding acquisition:** Ray Truant.

**Methodology:** Celeste Elisabeth Suart.

**Project administration:** Ray Truant.

**Supervision:** Ray Truant.

**Writing – original draft:** Celeste Elisabeth Suart.

**Writing – review & editing:** Katherine Jean Graham, Theresa Nowlan Suart, Ray Truant.

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
