## [Decision Letter · Decision Letter 0]

27 Apr 2020

PONE-D-20-05285

Development of a knowledge translation platform for ataxia: Impact on readers and volunteer contributors

PLOS ONE

Dear Prof. Truant,

Thank you for submitting your manuscript to PLOS ONE. After careful consideration, we feel that it has merit but does not fully meet PLOS ONE’s publication criteria as it currently stands. Therefore, we invite you to submit a revised version of the manuscript that addresses the points raised during the review process.

We would appreciate receiving your revised manuscript by Jun 11 2020 11:59PM. To enhance the reproducibility of your results, we recommend that if applicable you deposit your laboratory protocols in protocols.io, where a protocol can be assigned its own identifier (DOI) such that it can be cited independently in the future. For instructions see: http://journals.plos.org/plosone/s/submission-guidelines#loc-laboratory-protocols

We look forward to receiving your revised manuscript.

Kind regards,

Keith M. Harris, PhD

Academic Editor

PLOS ONE

Additional Editor Comments (if provided):

Dear Dr Truant,

Please closely read the comments by the reviewers. They offer helpful suggestions on improving this submission. In particular, further information on your project, details of collecting information, and activities by your group. would help enlighten this work for potential research consumers. After addressing these issues, we look forward to reviewing your updated manuscript.

Best wishes,

Keith

2. Please note an error on lines 144-145 as 6/19=32% and not 26%.

Reviewers' comments:

Reviewer's Responses to Questions

**Comments to the Author**

1. Is the manuscript technically sound, and do the data support the conclusions?

Reviewer #1: Yes

Reviewer #2: Partly

Reviewer #3: No

2. Has the statistical analysis been performed appropriately and rigorously? 

Reviewer #1: Yes

Reviewer #2: Yes

Reviewer #3: N/A

3. Have the authors made all data underlying the findings in their manuscript fully available?

Reviewer #1: Yes

Reviewer #2: Yes

Reviewer #3: No

4. Is the manuscript presented in an intelligible fashion and written in standard English?

Reviewer #1: Yes

Reviewer #2: Yes

Reviewer #3: Yes

5. Review Comments to the Author

Reviewer #1: The authors have developed a website called SCAsource to provide patients and their families with up-to-date health research information translated into lay terms. SCA stands for Spinocerebellar ataxia, which is a rare, fatal neurodegenerative disease. The online site also gives researchers the opportunity to develop skills in communicating their research to the public. The website was started in 2018 by a small group using their own funds. This article is a first attempt to evaluate the success of SCAsource. Figure 1 shows the distribution of 26,9000 views of SCAsource worldwide, indicating the website is being used. Surveys were sent to contributors and readers of SCAsource online with multiple choice quantitative questions and open-ended qualitative questions. A small sample size responded: 19 contributors (58% of those requested) and 36 users (63% of those requested). The authors are aware of the limitations of the small sample size, and including the qualitative responses was very informative. The authors summarize their results. Generally the contributors and respondents found value in the website. The contributors valued learning to communicate with the public. The readers valued access to research that is understandable and current. The authors suggest this is a good model for developing knowledge translation efforts for other rare diseases.

Overall the development of knowledge translation websites is a very valuable for scientists and for the people suffering from a disease. This effort will not only benefit the individuals involved, it will encourage support for research among the public. This is an important article to publish. As a graduate student I worked in the area of inhibitor design for HIV and volunteered at an AIDS hotline. I learned a lot from working with the hotline and it provided me with an emotional connection for the value of my research.

I have no major criticisms of the paper. I have a few comments that occurred to me while reading the article.

To the extent that the authors wish to promote other groups to emulate their successes, it would be helpful to provide a little more background about SCAsource and how it got started.

- line 68 states that the authors were inspired by discussions with their own patient group. Was SCAsource started by doctors who treat SCA patients? Is that a good model for starting a knowledge translation effort? I imagine the founders must have had some relationship with patients suffering from the disorder.

- What are the combination of skills needed to develop a website for other diseases? Did you need a computer scientist, a web designer, a survey of patients with SCA, contacts with the National Ataxia Foundation, …

Questions about the SCAsource knowledge-translation model:

- To what extent is this model filling a need to serve rare disease communities versus bridging researchers and patients of any disease community? Is this niche already being filled for more common disorders, eg cancer or diabetes?

- How many countries have an organization like the National Ataxia Foundation? Are you at all duplicating information that they have available? Does it make sense to be disseminating SCAsource through them instead of maintaining an independent website?

Questions about contributors:

- How do contributors find out about SCAsource? Is there a main SCA meeting that is attended by many researchers in the field? Is this an international meeting? Is there a journal that is read by many practitioners in the field?

- Where are your contributors from? Mostly Canada and the US, or other places in the world? Do they match the distribution of your readers (Figure 1b)?

- To what extent are the contributors in basic research vs clinical research? Have you been able to bridge the gap between those communities?

- How do contributors learn if their content is accessible to readers? Training seems to be through 6 pages of written documents for new volunteers.

Questions about readers:

- How big is the patient community with SCA? What percent of that community is using the website?

- Line 153 – you don’t specify what % of your readers responded to your survey (36/57 ~ 63%), though you say the % of your contributors that responded (line 143 - 58%).

Questions about content:

- Lines 236-9: The authors mention there was some increased interest among the readers to participate in clinical trials. Perhaps people interested in clinical trials are more likely to read research articles. To what extent is participation in clinical trials promoted in the content?

- I know that many people suffering from incurable diseases look to non-medical treatments, to mitigate suffering or even for cures. Does SCAsource address those concerns of readers? I don’t know if there is research in those areas.

- I like the idea of adding laboratory profiles as an article type. That will personalize the research for the SCA readers, which seems valuable.

Figure 1a. Do the colors mean anything in the bar graph?

If not, the colors are confusing because they are the same as in the key in part b.

- It’s interesting that the number of views go up and down, somewhat seasonally.

Is there a reason for that?

Reviewer #2: Thank you for the opportunity to review this paper which describes the development of a KT platform for ataxia. This platform - SCAsource is an online platform where peer reviewed research papers are translated into lay summaries. The goal was to make research more accessible and understandable to patients and families. A secondary goal was to provide opportunities for ataxia researchers to develop and hone their translational skills (to write lay summaries).

It is unclear if patients and families were part of the decision to develop this platform (which would be important) - or if they were passive recipients of the platform.

Two online surveys - developed for this study- were distributed (19 volunteers; 36 respondents). Descriptive statistics were conducted and grounded theory thematic analysis were conducted on the textual survey responses.

My main concerns are on the small sample of both volunteers and readers of the platform. This is a small number of participants to make the claims in the manuscript (eg the readers overwhelmingly appreciated the easy to understand content). I would encourage the writers to temper these claims in light of the sample size.

I would challenge the authors that they did not actually conduct grounded theory thematic analysis - rather, what they describe is more aligned with Qualitative Description. There was no mention of steps to address rigor in their qualitative analysis.

I would suggest that the Discussion section needs to be revised. There are many different initiatives that could be used to situate these findings against (e.g., Cochrane Collaboration has been doing lay summaries for quite some time). The discussion needs to be enhanced.

Reviewer #3: The manuscript describes an online platform that was launched in September 2018, with the goal of making ataxia research more accessible and understandable to patients and families. A secondary goal was to provide opportunities for ataxia researchers to develop and hone their knowledge translation skills, which would then improve the quality of patient communication in the ataxia community. The study (unfunded) measured the impact of SCAsource on its readers and volunteer contributors after one year of activity.

Background

Pg 4 - The notion of a snapshot presented here sounds like a summary of a summary. I would argue that summaries are bound by the parameters of the research study; they summarize the study in simpler terms for the lay reader. Snapshots, however, should do more than summarize; they must offer new information in the context of the informational needs of the reader. Snapshots should convey why the aim of the study and its’ findings are important to the reader; what are the implications and potential benefits of the study for the reader? In this way, the scientific work is tailored to the needs of the reader and are key to ensuring lay readers benefit from the deliverable.

Pg 5 - It appears the expectation was that contributors would improve their lay writing skills simply by being given the opportunity to do so? Yet, plain language writing is a teachable skill; how did you intervene to teach them? There is mention that resources were shared with contributors, but these are not critically appraised for quality, nor are they shared in the manuscript. It also seems rather naïve to assume that written documentation can suffice to train writing skills in the absence of trial /feedback, and experience.

Pg 5, line 97-98 – this sentence belongs in the conclusion.

Overall, linkage with the KT literature is fairly weak in this paper and contributes to the impression that the project is atheoretical and not guided by the knowledge base in dissemination.

Methods

1. It seems curious to me that this work was waived by REB; on what grounds? You are collecting human data and publishing. Please describe.

2. There were not additional data files attached to the submission, i.e., survey protocols, teaching materials and examples provided to the contributors.

3. The survey methodology could be improved upon based on current methods of this kind published in the literature. For instance, methods that allow for the invitation of content feedback linked directly on the webpage; see the work of Pierre Pluye and McGill University on the Information Assessment Method.

4. Pg 7 – Web analytics were reportedly collected but don’t figure at all in the results or discussion.

5. Pg 7-8 –It's hard to ascribe any meaning to the contributor characteristics without further information about how they are recruited, supported, trained, remunerated, etc.

6. p9 – The use of survey format with open-ended options demonstrates a lack of patient-centeredness. Presumably and in fact, intentionally, the limitations of the respondents with respect to response options should have been anticipated and another modality of feedback provided; phone, for instance, or multiple choice rather than open ended questions, or open-ended questions that could be voice recorded.

7. P9 - There is no information about reader demographics - age, sex, patient status, etc - and thus their responses are not contextualized. Without this information, it's hard to understand why they may have said /rated how they did so. This is another example of who the project doesn't seem all that well thought out in terms of user-perspective, KT training and quality of dissemination products.

8. P16 - How are summaries/snapshots vetted prior to publication? What are the standards of quality and how are they assessed? They may like doing it but are they any good at it? The latter seems to be the more important question, in addition to whether readers benefit from them, and how.

9. Since your sample numbers are small, the use of percentages has the effect of overamplifying. Please convey the n as well as the % so as not to over emphasize the findings.

10. All of the survey questions were very researcher oriented; focused on how the readership can support research rather than how the research, when simply communicated, can benefit the reader/patient. This seems rather self-serving and not at all in the spirit of KT.

11. How were the content formats decided upon? by whom? how do you know this is what readers want?

12. P 19, line 248/9 - Again, as mentioned above, the researchers place importance on how clearly a scientific concept or finding can be communicated to a lay reader. What about focusing on whether the summary can inform and benefit the reader? This would seem to be more laudable and purposeful goal of effective dissemination.

13. P 23, line 276 - this is really curious because most lay people have no or little interest in research papers, not only because they are behind pay-walls and filled with jargon, but more importantly because they do not address the meaning of the research from the readers perspective. This resource does nothing to address this.

14. There is no linkage to the literature on what other laypersons seek from research summaries; what type of information and format they prefer. The paper would be strengthened by a much tighter connection to the literature.

Discussion

15. p 24, line 302 - Depends how you interpret highly positive; they don't seem to report many benefits, so this statement would seem to be unsubstantiated.

16. Dissemination of health information on the web isn't a model; it's a dissemination strategy. Models simplify a process of translating evidence into practice. Models provide the key steps to plan for and undertake when operationalizing a D&I initiative. You don't have a model; you have a platform for dissemination.

17. It is odd for contributors to practice something for which they have received little instruction or that is not informed by an evidence-informed framework. There are published snapshot frameworks and frameworks for effect dissemination, but this paper does not use them to inform the work, rendering the exercise atheoretical and yielding little of value to KT science or the readership.

I recognize the aim of this website is to make research accessible to knowledge users, and this is important. That said, there is little depth with respect of KT evidence or methodological approach or capacity building/training of contributors/academic faculty, and thus the manuscript does not provide novel information that moves the field forward.

6. PLOS authors have the option to publish the peer review history of their article (what does this mean?). If published, this will include your full peer review and any attached files.

Reviewer #1: No

Reviewer #2: No

Reviewer #3: Yes: Melanie Barwick

---

## [Author Response · Author response to Decision Letter 0]

9 Jul 2020

We would like to thank all three reviewers for their expertise and comments, especially during this period of pandemic response. Based on these comments, we significantly revised the manuscript writing and believe we have a much-improved manuscript. This project has been a quantified success and a potential template for rare diseases knowledge translation, which by nature, has to rely on a low cost, volunteer-driven approach without need for computer IT expertise. 

Below are point-by-point responses.

PLOS One SCAsource Manuscript Reviews

Reviewer 1

To the extent that the authors wish to promote other groups to emulate their successes, it would be helpful to provide a little more background about SCAsource and how it got started.

- line 68 states that the authors were inspired by discussions with their own patient group. Was SCAsource started by doctors who treat SCA patients? Is that a good model for starting a knowledge translation effort? I imagine the founders must have had some relationship with patients suffering from the disorder.

● SCAsource was started by a group of graduate student and postdoctoral fellows following discussions with patients and family members at the 2018 NAF Ataxia investigators Meeting in Philadelphia. Part of this meeting involves a knowledge translation poster session, where we received feedback from individuals they wished there was “an HDBuzz for ataxia” – this and the success of HDBuzz in the Huntington’s Disease community led us to follow this model. Further clarification had been added to the text to reflect this origin.

- What are the combination of skills needed to develop a website for other diseases? Did you need a computer scientist, a web designer, a survey of patients with SCA, contacts with the National Ataxia Foundation, …

● We used WordPress, an online content management system which allowed for the creation of a professional website by persons with limited web design experience, in order to reduce startup costs to less than 500USD. Thus, we didn’t need to spend money on a computer scientist or web designer to set up the website. We didn’t have the financial or technical resources to conduct a patient survey at the time, so most feedback was received informally through one-on-one conversations or by email. The national ataxia foundation did provide some support through including information about SCAsource in their e-newsletter and social networks, however, they wanted to see proof of impact (the results of this study) before supporting the website financially. Writing guidelines and training was developed by volunteers who had previous knowledge translation training with other organizations. Further detail has been added to describe the set-up of the SCAsource website in order for other group to replicate this method [Line 88-97]

Questions about the SCAsource knowledge-translation model:

- To what extent is this model filling a need to serve rare disease communities versus bridging researchers and patients of any disease community? Is this niche already being filled for more common disorders, eg cancer or diabetes?

● This reviewer raises a good point, that such a set-up could be used to bridge researchers and patients in any disease community, as opposed to only rare diseases. We initially proposed this model to rare disease interest groups for two reasons. More common disorders tend to have already established KT efforts, while rare disease organizations typically receive less funding than more common disorders. Due to our set-up being a low-cost initiative, we thought this may better suit the needs of rare disease organizations. We have clarified this in the discussion of the manuscript [Line 417-426]

- How many countries have an organization like the National Ataxia Foundation? Are you at all duplicating information that they have available? Does it make sense to be disseminating SCAsource through them instead of maintaining an independent website?

● There are numerous ataxia organizations and charities around the world. Although some of these organizations would have sporadic knowledge translation efforts, there was not consistent engagement in these activities. As we were filling this gap, both the National Ataxia Foundation and other international organizations have started using our content. As there is no one worldwide organization for ataxia, the decision was made to have an independent website in an attempt to bridge some of these geographical divides in audience. Further clarification has been added to the manuscript to reflect this. [Line 90-92]

Questions about contributors:

- How do contributors find out about SCAsource? Is there a main SCA meeting that is attended by many researchers in the field? Is this an international meeting? Is there a journal that is read by many practitioners in the field?

● The text has been updated to reflect this information. [Line 98-100] We have not used journal advertisements at this time – primarily due to lack of funding. This is something we hope to explore in the future. 

- Where are your contributors from? Mostly Canada and the US, or other places in the world? Do they match the distribution of your readers (Figure 1b)?

● The text has been updated to reflect this information. [Line 100-101] The majority of volunteers are from the United States, followed by Canada, which reflected trends seen in 1b. However, the remainders of volunteers are largely from Europe, which is not entirely reflective of our readership. A point to this effect has been added to the discussion [Line 370-373].

- To what extent are the contributors in basic research vs clinical research? Have you been able to bridge the gap between those communities?

● The text has been updated to reflect this information. [Line 101-102] The majority of contributors are basic science researchers, but our cohort of clinical researchers is expanding. Unfortunately our plans to expanded our clinician-researcher pool of volunteers by through recruitment at a conference in March was disrupted due to COVID-19

- How do contributors learn if their content is accessible to readers? Training seems to be through 6 pages of written documents for new volunteers.

● The text has been updated to reflect this information. [Line 103-109] Training is accomplished through two steps – assigned reading to learn the SCAsource article standards and basic theory. Most growth and development occurs during the writing process, when less experienced writers are paired with more experienced editors. Through an iterative editing process, the volunteers can hone their lay summary writing skills. Later on, they can mentor new volunteers in turn. 

Questions about readers:

- How big is the patient community with SCA? What percent of that community is using the website?

● There are no accurate worldwide estimates of the SCA patient community. Internal numbers from the National Ataxia Foundation estimate 150,000 individuals with ataxia in the United States. With 11,500 unique IP addresses visiting SCAsource from September 2018 to January 2020, with 54.7% of SCAsource visitors originating in the USA, we could estimate 4.2% of US SCA patients using the website. However, this would not take into account multiple patients using the same IP address, or one patient using multiple IP addresses. Due to the uncertainty of these numbers and multiple assumptions required to make the estimate, we are not able provide an accurate estimate.

- Line 153 – you don’t specify what % of your readers responded to your survey (36/57 ~ 63%), though you say the % of your contributors that responded (line 143 - 58%).

● As we know the total number of contributors to SCAsource, we chose to display a percentage breakdown of respondents to reflect the sample vs total population. Although we know how many individuals signed up to receive SCAsource email updates (57), this number does not reflect who reads the website frequently, but does not wish to receive emails. Due to the lack of an accurate estimate of total readership, we did not include a percentage breakdown of reader respondents vs total population.

● We have added to our limitations section within the discussion a section on the sample size of the reader respondents relative to total views (as a stand in for total population) to help clarify this point [Line 448-451].

Questions about content:

- Lines 236-9: The authors mention there was some increased interest among the readers to participate in clinical trials. Perhaps people interested in clinical trials are more likely to read research articles. To what extent is participation in clinical trials promoted in the content?

● SCAsource does not actively promote clinical trial registration and participation in our content. Due to reader interest and suggestions, we have covered information about how clinical trials are run and analysed. If relevant to the article summary, we also explain if a discovery is about to enter clinical trials or the published results of clinical trials. 

● The reviewer is raising a good point, that patients intrinsically more interested in clinical trial participation could be more likely to seek out research information. We could not find literature about this correlation. However, one documented barrier to clinical trial participation is inadequate information about the trial process, while conversely information from trusted sources is cited as a motivating factor for clinical trial participation. Further clarification has been added to the manuscript to reflect this. [Line 294-405]

- I know that many people suffering from incurable diseases look to non-medical treatments, to mitigate suffering or even for cures. Does SCAsource address those concerns of readers? I don’t know if there is research in those areas.

● This does happen within ataxia, with patients seeking non-medical treatments that have varying levels of scientific backing. We mainly address these concerns through our Snapshot article series. For example, over the summer there was a spike in search terms along the lines of “stem cell injections cure ataxia” directing traffic to our website. In response to this, we published snapshots on stem cells to clarify what they can (and cannot) do. Through these Snapshots we are attempting to debunk some misconceptions that are out there. 

- I like the idea of adding laboratory profiles as an article type. That will personalize the research for the SCA readers, which seems valuable.

● We would like to thank the reviewer for this positive feedback. This initiative has been put on hold due to the COVID-19 disruption, but we hope to begin it again soon.

Figure 1a. Do the colors mean anything in the bar graph?

If not, the colors are confusing because they are the same as in the key in part b.

● The colours in figure 1a have been adjusted to all be one colour to reduce potential confusion. 

- It’s interesting that the number of views go up and down, somewhat seasonally.

Is there a reason for that?

● Seasonality of web page views has been extensively studied in business, with SCAsource following a similar trend to other non-commercial business, where there is a decrease of view over the holiday season (November-December). SCAsource also sees a consistent spike of views in October. We think this may be due to increased awareness of the website following international ataxia awareness day (September 25). However, this is conjecture on our part. A point to this effect has been added [Line 365-368]

Reviewer 2

It is unclear if patients and families were part of the decision to develop this platform (which would be important) - or if they were passive recipients of the platform.

- Further detail has been added to describe how the idea for SCAsource came from discussions with ataxia patients and family members at the 2018 NAF Ataxia investigators Meeting in Philadelphia. Part of this meeting involves a knowledge translation poster session, where we received feedback from individuals they wished there was “an HDBuzz for ataxia” [Line 73-74]

- We have further clarified how information reader feedback has shaped SCAsource’s content, including the launch of the “Snapshot” article type and recommendations of topics to include [Line 116-118]

- We have not had the opportunity to collect formal feedback from readers until this study. However, the results of these surveys has allowed us to build further inroad with Ataxia charities, who have the expertise to grow to the point where we can have a lay person advisory committee, which is a long term goal of our initiative. 

My main concerns are on the small sample of both volunteers and readers of the platform. This is a small number of participants to make the claims in the manuscript (eg the readers overwhelmingly appreciated the easy to understand content). I would encourage the writers to temper these claims in light of the sample size.

● We have tempered claim descriptions within the manuscript to reflect the small sample size, in addition to including further information in the Limitations section of the study with regards to the small sample size [Line 448-456].

I would challenge the authors that they did not actually conduct grounded theory thematic analysis - rather, what they describe is more aligned with Qualitative Description. There was no mention of steps to address rigor in their qualitative analysis.

- We would like to thank this reviewer for bringing Qualitative Description to our attention. We were not familiar with the nursing and midwifery literature on Qualitative Description as a method prior to this project, but it would have been a good fit for the analysis goals of this project. This information will be very helpful to us as we move forward on other projects. As we were not aware of Qualitative Description, we chose to use the coding methodology described in grounded theory as it seemed to offer the most objective method to derive meaning from the survey data. That being said, we did not conduct a full grounded theory analysis as we did not conduct axial coding or develop an overarching theory from our data set (our sample size and total population is too small to do this). We have added additional clarification about this [Line 174]. Steps taken to ensure rigour of qualitative analysis have been further emphasized in the manuscript [Line 181-188].

I would suggest that the Discussion section needs to be revised. There are many different initiatives that could be used to situate these findings against (e.g., Cochrane Collaboration has been doing lay summaries for quite some time). The discussion needs to be enhanced.

- The discussion has been enhanced with further connection to the knowledge translation literature, including barriers to engaging in KT initiatives and why popular KT models may not fit the rare disease context. [Line 417-442]

o Rare disease organizations have previously identified knowledge translation as a needed area of improvement. More common disease tend to have more established knowledge translation initiatives. One potential barrier to explain this discrepancy is cost, as rare diseases typically on the whole receive less research funding. We propose our set-up outlined in this manuscript reduces these barriers to entry due to lower costs. 

o Multiple knowledge translation models have been documented in the literature, with overlap allowing for key themes of iterative process, stakeholder engagement, consulting multiple viewpoints. However, other groups have noted that these existing models do not address specific issues relevant to the rare disease context (focusing on improved survival and quality of life when rare disease treatment tends to have incremental effects, policy makers unlikely to make significant changes for small population sizes, traditional markers of evidence quality as many rare diseases do not have enough patients to support RCTs or meta-analyses, etc.).

o For many SCAsource readers, the research being done right now will not directly impact their health care decisions right now.The majority of research that SCAsource is asked to cover is either pre-clinical or in early clinical trials. Rather than traditional outcomes measured by many KT models, learning more about these findings may influence a reader’s choice to participate in research, or provide hope that new discoveries are being made. Although different in nature then other KT initiatives, we argue that these outcomes are important for this patient population where currently there are no preventative treatments available. 

- We have also added a section discussing whether reading SCAsource materials make people more likely to participate in clinical research, or if people already interested in clinical research research are more likely to seek out material like what is covered by SCAsource. We included this section to temper our claims in the manuscript by acknowledging both of these possibilities, while tying it to previous findings in the literature on this topic. [Line 394-405]

Reviewer 3

Background

Pg 4 - The notion of a snapshot presented here sounds like a summary of a summary. I would argue that summaries are bound by the parameters of the research study; they summarize the study in simpler terms for the lay reader. Snapshots, however, should do more than summarize; they must offer new information in the context of the informational needs of the reader. Snapshots should convey why the aim of the study and its findings are important to the reader; what are the implications and potential benefits of the study for the reader? In this way, the scientific work is tailored to the needs of the reader and are key to ensuring lay readers benefit from the deliverable.

- We believe there has been a misunderstanding in our use of terms. We have now learned that a style of lay article called “ResearchSnapshots” has been described by researchers at York University. We do not use the term “snapshot” in the same sense as this group, as their description of this article style more closely aligns with our “Summary” style articles.

- We use the term “Snapshot” in the colloquial sense of a brief look or overview, a reference to conciseness of this article type. This style of article came from informal feedback from readers that they wanted more explanations of key foundational concepts that are touched upon in Summary-style articles, such as techniques, cell types, or similar background information. Further clarification has been added to the manuscript to reflect this. [Line 115-121]

- Reviewer three’s suggestions (aim of the study, findings, implications, and potential benefits to the reader) are covered in the Summary style of article, which focuses on conveying key messages from one (or sometimes multiple) scientific research articles. Further clarification has been added to the manuscript to reflect this. [Line 111-115]

Pg 5 - It appears the expectation was that contributors would improve their lay writing skills simply by being given the opportunity to do so? Yet, plain language writing is a teachable skill; how did you intervene to teach them? There is mention that resources were shared with contributors, but these are not critically appraised for quality, nor are they shared in the manuscript. It also seems rather naïve to assume that written documentation can suffice to train writing skills in the absence of trial /feedback, and experience.

- We agree with reviewer three that written documentation alone is not sufficient to teach knowledge transition skills. The written resource serves as a tool for volunteer contributors to learn the procedures SCAsource has to facilitate the writing process, the minimum quality standards that articles must meet to be published, as well as background literature on Scientific writing including guidelines by Joselita Salita. This gives a new writer a baseline level of background knowledge. The skill development occurs during the facilitated editing process where less experienced volunteers received asynchronous mentorship from the editor. Further clarification has been added to the manuscript to reflect this. [Line 103-109, 123-132]

- We did not include these with the manuscript, as similar KT manuscripts published on PLOS ONE do not include similar teaching materials. As the main goal of this paper is to communicate a low-budget method of launching a KT website, we have included a statement [Line 474-475] that these materials will be available upon request. This will allow us to track other research groups which use a similar dissemination strategy. 

Pg 5, line 97-98 – this sentence belongs in the conclusion.

- This sentence has been removed from the introduction as recommended

Overall, linkage with the KT literature is fairly weak in this paper and contributes to the impression that the project is atheoretical and not guided by the knowledge base in dissemination.

- We have added additional connection to the theoretical KT literature (Knowledge to action model by Graham and colleagues) to the introduction. Our focus remains primarily on the lay summary literature, as this subsection of KT is the most relevant to our context. As no one KT theoretical framework especially espouses the use of lay summaries, we thus chose to ground the introduction in practical research that has been done in lay summaries.

Methods

1. It seems curious to me that this work was waived by REB; on what grounds? You are collecting human data and publishing. Please describe.

- This project was evaluated by the Hamilton Integrated Research Ethics Board (see attached letters) who judged this work to be a quality assurance project exempt from ethics review. The publication of this data, since it was anonymous, was considered to be secondary use of quality assurance data. We have included further clarification to this effect in the manuscript [Line 141-143]

2. There were not additional data files attached to the submission, i.e., survey protocols, teaching materials and examples provided to the contributors.

- We submitted the manuscript from bioRxiv a pre-print directly to PLOS ONE, during this process we were unable to attach additional documentation. In this resubmission we have included copies of the survey protocol, codebook (embedded in the manuscript text), and ethics exemption letters.

3. The survey methodology could be improved upon based on current methods of this kind published in the literature. For instance, methods that allow for the invitation of content feedback linked directly on the webpage; see the work of Pierre Pluye and McGill University on the Information Assessment Method.

- We would like to thank reviewer three for highlighting the work of Pierre Pluye and colleagues. We will keep this in mind for future analysis of SCAsource. For the current study, we did include an invitation to participate in the survey to give feedback directly on the SCAsource website. We are looking into embedding a feedback function on all articles to give real-time feedback, however, these efforts have been prevented by our current financial limitations. 

4. Pg 7 – Web analytics were reportedly collected but don’t figure at all in the results or discussion.

- This figure has been moved to the result section. In the discussion, we have added text comparing the location of SCAsource readers to the location of contributors, as well of seasonal patterns of views [Line 191-198]

5. Pg 7-8 –It's hard to ascribe any meaning to the contributor characteristics without further information about how they are recruited, supported, trained, remunerated, etc.

- Further background on volunteer contributors has been included in the introduction [Line 98-109]. This includes recruitment practices (primarily word-of-mouth), training documents, and mentorship received from the writing process. As we have been working on a limited budget, all contributors are volunteers. They receive no payment. We hope to provide honoraria in the future when we secure more financial support. 

6. p9 – The use of survey format with open-ended options demonstrates a lack of patient-centeredness. Presumably and in fact, intentionally, the limitations of the respondents with respect to response options should have been anticipated and another modality of feedback provided; phone, for instance, or multiple choice rather than open ended questions, or open-ended questions that could be voice recorded.

- We agree with reviewer three that the mode of study (online survey) was not ideal for parts of this ataxia patient population, especially those in later stages of disease. A discussion of these limitation is included in the results [Line 211-219] and discussion [Line 444-449].

- We were constrained to the use of online surveys for two reasons; limited financial resources and the world-wide distribution of SCAsource readers. We did not have the money or person-power to be able to offer telephone interviews for national, let alone international calling. We hope to be able to conduct further analysis of SCAsource in the future, with proper financial support, in order to make the data collection process more accessible. 

- When designing the survey tool for both contributors and readers, we were intentional with our use of both multiple choice, Likert-type, and open-ended questions. As this was the first larger-scale attempt to receive feedback from readers and volunteers, we did not want to presume or put our own biases as researchers on potential likes and dislikes of SCAsource. For this reason, these questions were left open-ended. Questions where we could reasonably predict potential responses (such as characteristic information) were styled as multiple-choice options to decrease as many barriers as possible.

7. P9 - There is no information about reader demographics - age, sex, patient status, etc - and thus their responses are not contextualized. Without this information, it's hard to understand why they may have said /rated how they did so. This is another example of who the project doesn't seem all that well thought out in terms of user-perspective, KT training and quality of dissemination products.

- For this first analysis of feedback from readers, we were looking for four key variables to contextualize responses: frequency of reading SCAsource content, how frequently they look for ataxia information overall, sources of ataxia information, and where they find SCAsource content. With this information we were able to understand trends in why readers responded in certain ways. 

- In terms of collecting other demographic data (age, sex, gene status), we ran into two barriers. First, due to potential respondents being from 100+ countries, we ran into the issue of having to navigate the variety of privacy laws and regulations. Due to limited resources, this would not have been feasible for our team. For this reason, when choosing between collecting some key information or none at all, we chose to collect the information described in the first bullet point, which would have the greatest impact on readers’ interaction with the site. Second, we hesitated to inquire about patient gene or disease progression status due to inconsistency of genetic non-discrimination policies (many forms of ataxia being genetic). One issue that many patients and families face when other learn of gene-positive status is discrimination from insurance, work, and education organizations. Due to this context, patients often only disclose gene status to their healthcare team and for related medical studies. As we are not a medical study, we did not want to discourage participation by requiring gene status disclosure. 

8. P16 - How are summaries/snapshots vetted prior to publication? What are the standards of quality and how are they assessed? They may like doing it but are they any good at it? The latter seems to be the more important question, in addition to whether readers benefit from them, and how.

- Further information about the editing process and quality standards has been added [Line 123-132]. SCAsource contributors follow a month-long article writing and editing process: Two weeks for first drafts of articles, one week for editor review, then one week to resubmit articles. If they do not meet minimum requirements, article are place back into the next writing/editing cycle. Requirements include style and content guides described by Joselita Salita. Articles must also score a minimum of 80% of the “suitability for general audience” measure by De-Jargonizer, an online automated jargon identification program. Articles are then sent for copy-editing and formatting prior to publication.

- Our aim of this study was to formally document if and how readers benefited from reading SCAsource content. 

9. Since your sample numbers are small, the use of percentages has the effect of overamplifying. Please convey the n as well as the % so as not to over emphasize the findings.

- The n values have been added for percentages relating to the reader and contributor data.

10. All of the survey questions were very researcher oriented; focused on how the readership can support research rather than how the research, when simply communicated, can benefit the reader/patient. This seems rather self-serving and not at all in the spirit of KT.

- Survey questions related to contributors are more researcher oriented, due to contributors being researchers themselves. Questions related to readers were more oriented towards patients and families, including which article types were most helpful, understanding of & connection to ongoing research (an interest of this population which we seek to serve), and if they trusted SCAsource as a source of information. 

- Open-ended questions about likes and dislikes about the SCAsource were included for both populations in order to gain insight into the values from these populations, rather then imposing our own self-interests upon these groups. 

- The single question of interest in participating in clinical trials was included following informal discussions with potential funding sources, as they wanted to know this information. This can be described as self-serving, as our long term goal is to find funding sources to be able to continue KT efforts through SCAsource. Without this, we would need to stop all knowledge translation activities. 

11. How were the content formats decided upon? by whom? how do you know this is what readers want?

- Information on content formats has been expanded upon in the introduction [Line 110-122]. The Summary article type was modeled after the types of article posted on HDBuzz. We kept this article type due to the inspiration for SCAsource being a “HDBuzz-like” website for SCAs and Ataxia (as requested by patients and families). The Snapshot article type was launched roughly six months after SCAsource went live, following informal feedback from readers to the site that they wanted more information on background concepts in science. 

- This study was the first instance to collect formal feedback from readers on if these content styles were helpful to them. Our findings support that they want both of these article styles. 

12. P 19, line 248/9 - Again, as mentioned above, the researchers place importance on how clearly a scientific concept or finding can be communicated to a lay reader. What about focusing on whether the summary can inform and benefit the reader? This would seem to be more laudable and purposeful goal of effective dissemination.

- The description on line 306/307 (Previously line 248/9 prior to edits) focuses on highlighting the differences between the Summary and Snapshot style of articles. “Cleary and concisely” was used as Snapshots are much shorter and more succinct than Summaries. As described earlier in this response, this was requested informally from the reader community, so that they could have a better understanding of background knowledge used in Summaries. 

- All Snapshots include information as to how this knowledge benefits either the research process or directly to ataxia patients, as this is a fundamental part of KT work. Often topics discussed in summaries do not have a direct benefit to readers, in these cases we make connections as two how the topic benefits the research process, which in turn could benefit the reader. 

13. P 23, line 276 - this is really curious because most lay people have no or little interest in research papers, not only because they are behind pay-walls and filled with jargon, but more importantly because they do not address the meaning of the research from the readers perspective. This resource does nothing to address this.

- We agree with reviewer three that there are multiple barriers for lay person accessing primary literature and research, especially that many researchers do not make the connection to how patients can benefit from findings. One of the goals of KT being done by SCAsource is to make these connections explicit. 

- We respectfully disagree that most lay persons do not have interest in research papers. It is possible that the ataxia community is more engaged with ongoing research than other patient populations due to the nature of the disease. Most types of ataxia, including SCAs, are progressive, fatal diseases for which we do not have any form of preventative treatment. We are, however, entering an era where more and more treatments are entering pre-clinical and clinical trials. Patients are turning to the literature for hope. We have added a sentence to the manuscript connecting this finding to past literature showing patient interest in research literature [Line 336-337].

14. There is no linkage to the literature on what other laypersons seek from research summaries; what type of information and format they prefer. The paper would be strengthened by a much tighter connection to the literature.

- Additional literature context has been added to the introduction highlighting why certain article formats were selected for SCAsource [Line 110-122]. This includes the inverted pyramid model and best practices outlined by Joselita Salita, as well as work by Barnfield and colleagues during the COGFAST study, where they asked members of the public on what format and content they prefer in lay summaries.

Discussion

15. p 24, line 302 - Depends how you interpret highly positive; they don't seem to report many benefits, so this statement would seem to be unsubstantiated.

- We have removed the word “highly” to temper our claims given the small sample size from the reader population.

- We respectfully disagree with the description the statement “Overall, both groups reported a positive evaluation of SCAsource” is unsubstantiated as both groups reported positive evaluation of the platform in both quantitative (Fig 2-4) and qualitative data (Table 3 and 4). 

16. Dissemination of health information on the web isn't a model; it's a dissemination strategy. Models simplify a process of translating evidence into practice. Models provide the key steps to plan for and undertake when operationalizing a D&I initiative. You don't have a model; you have a platform for dissemination.

- We have modified the text to say “dissemination strategy” instead of “model”. 

17. It is odd for contributors to practice something for which they have received little instruction or that is not informed by an evidence-informed framework. There are published snapshot frameworks and frameworks for effect dissemination, but this paper does not use them to inform the work, rendering the exercise atheoretical and yielding little of value to KT science or the readership.

- We respectfully disagree with reviewer three on this comment. When designing SCAsource we use the framework previously outlined by HDBuzz. This choice was made primarily due to patients requesting a similar platform style for ataxia, but also that HDBuzz has had documented successful engagement with its readership. 

- To the point that contributors have received little instruction in KT, this is a gap that SCAsource is trying to fill. Due to being research in a rare disease field, there is limited research funding and professional development. This has led to a gap in training for researchers in this area. Many external forms of formal KT training are cost-prohibitive. Thus, our training comes primarily through reading the KT literature. We do not claim that our current training is perfect, it is highlighted as an area of improvement in this study. Without the results of this study demonstrating impact, we would never be able to gain funding in order to put more extensive training in place. 

- 

Our aim with this paper is not to further KT theory, as there are more resourced and experienced research groups which do this. Our aim was more practical in nature, to document and outline a low-cost dissemination strategy which other groups with funding concerns could replicate. For rare disease groups who have limited funding resources, cost can be prohibitive to even beginning the simplest of KT efforts. In this paper we outline a way to begin a KT platform for less than 500USD and demonstrate positive self-reported outcomes for both contributors and readers. We argue that we are providing value by (1) replicating past findings by HDBuzz focusing on patients using a similar dissemination initiative, (2) furthering this knowledge to include the perspective of contributors, and (3) outlining the practical steps that other groups could take to make similar platforms.

---

## [Decision Letter · Decision Letter 1]

19 Aug 2020

Development of a knowledge translation platform for ataxia: Impact on readers and volunteer contributors

PONE-D-20-05285R1

Dear Dr. Truant,

We’re pleased to inform you that your manuscript has been judged scientifically suitable for publication and will be formally accepted for publication once it meets all outstanding technical requirements.

Kind regards,

Keith M. Harris, PhD

Academic Editor

PLOS ONE

Additional Editor Comments (optional):

Dear Authors. Congratulations on your work and this positive outcome. Both reviewers were appreciative of your work, including your revisions. They also both suggested a few additional minor edits. I encourage you to finalize those few edits before we move forward and publish your work.

Reviewers' comments:

Reviewer's Responses to Questions

**Comments to the Author**

1. If the authors have adequately addressed your comments raised in a previous round of review and you feel that this manuscript is now acceptable for publication, you may indicate that here to bypass the “Comments to the Author” section, enter your conflict of interest statement in the “Confidential to Editor” section, and submit your "Accept" recommendation.

Reviewer #1: (No Response)

Reviewer #3: All comments have been addressed

2. Is the manuscript technically sound, and do the data support the conclusions?

Reviewer #1: Yes

Reviewer #3: Yes

3. Has the statistical analysis been performed appropriately and rigorously? 

Reviewer #1: N/A

Reviewer #3: Yes

4. Have the authors made all data underlying the findings in their manuscript fully available?

Reviewer #1: Yes

Reviewer #3: Yes

5. Is the manuscript presented in an intelligible fashion and written in standard English?

Reviewer #1: Yes

Reviewer #3: Yes

6. Review Comments to the Author

Reviewer #1: The paper is much improved, and I believe should be published.

The authors addressed all of my questions – in the article and through their detailed responses to my questions. I think the paper clearly indicates the limits of the current study, particularly the small sample size and whether their sample is representative of the community as a whole. But the responses to their survey provide an important first look at the value of their website. The authors are also clear that their initial evaluation will allow them to expand their effort by gaining support and funding from the ataxia community.

I have read the responses of the other reviewers to the first draft of the article. I am a scientist not versed in the literature of assessment. While I think it is valuable to criticize the methodology of the current study, it is important that it does not detract from the success and efforts represented by this project.

The one question I still had when I read the paper was how big the SCA research and affected communities are. The authors addressed the uncertainty of the affected community size in their author response (estimated 150,000 with SCA in the US). I think it is worth providing some idea of the community size in the paper because it justifies the need to expand their efforts. It also may be important for comparison by other groups interested in emulating this effort.

SCAsource is a valuable project that will have general interest.

First, the scientific community needs to foster communication between scientists and the public. This is clear from the skepticism by the public to medical advice and vaccines to Covid-19. Greater communication is important for scientists to understand how their work affects the community, but also for the community to develop trust in science.

I think this project has benefits beyond developing communication skills for scientists. Many scientists value the contribution that their work makes to society. As a researcher I have found that engaging with the community is rewarding and motivating. I think this is especially true for graduate students and postdocs. This point is addressed in the paper.

There is also a clear desire by people with SCA, and any disorder, to understand the symptoms, cause and possible treatments. This is an important trend in the internet age. Providing accurate information to the public is important since there is so much misinformation on the internet.

Grammatical corrections:

Line 48: the word “cation” needs to be corrected

Line 103-104: modify the sentence “New volunteers are giving….” I think they are given a training guide and document guidelines for the two different document types. You mention the editing feedback in a later sentence.

Line 187: change “vias” to “bias”

Line 316: change “ewer” to “fewer”

Reviewer #3: Thank you for the opportunity to read this revision. The paper describes a novel effort to improve access to empirical work in Ataxia while building capacity for KT among researchers - both are needed. There is a good model for knowledge sharing in other disease areas. Overall, I think the authors did a thorough job of responding to the reviewer comments. In reading the revision, I found a few more required revisions.

• Page 3. Type/part of sentence is missing. “Knowledge-to-Action, outlines both cycles of knowledge creation and knowledge application [5,6]cation cycle involves adapting knowledge to the local context of users, as well as identifying barriers to using and accessing this knowledge [5].

• Page 4 –typo - New volunteers are giving a training guide on how to write effective lay summaries…

• Page 5 – perhaps you could say something about this method, otherwise the reader has to go to the Salita paper: “Summaries follow the inverted pyramid structure and best lay summary practices described by Salita [15].”

• Page 8; change the verb ‘was’ to ‘were’; data is plural. “Once data was collected, survey response data was formatted

• Page 21 – type. “ewer readers rated Snapshots”

• I still maintain that the term ‘writer’ is clearer that the term ‘volunteers’. I understand the writers are volunteers, but their role on the site is as writers, and it flows better to distinguish between readers and writers.

• Page 27 – suggest ‘of’ instead of ‘by’ in this sentence, “Another assumption by knowledge translation models. Moreover, I would say that another assumption is that the knowledge being translated has immediate benefits for the reader rather than specifying clinical or instrumental benefits. Such benefits might include increased awareness, knowledge, or support in decision-making.

7. PLOS authors have the option to publish the peer review history of their article (what does this mean?). If published, this will include your full peer review and any attached files.

Reviewer #1: No

Reviewer #3: **Yes: **Melanie Barwick

---

## [Editor Report · Acceptance letter]

20 Aug 2020

PONE-D-20-05285R1 

Development of a knowledge translation platform for ataxia: Impact on readers and volunteer contributors 

Dear Dr. Truant:

I'm pleased to inform you that your manuscript has been deemed suitable for publication in PLOS ONE. Congratulations! Your manuscript is now with our production department. 

Kind regards, 

on behalf of

Dr. Keith M. Harris 

Academic Editor

PLOS ONE